# ER-Localized PIN Carriers: Regulators of Intracellular Auxin Homeostasis

**DOI:** 10.3390/plants9111527

**Published:** 2020-11-10

**Authors:** Nayyer Abdollahi Sisi, Kamil Růžička

**Affiliations:** 1Laboratory of Hormonal Regulations in Plants, Institute of Experimental Botany, Czech Academy of Sciences, 16502 Prague, Czech Republic; Abdollahi@ueb.cas.cz; 2Department of Experimental Plant Biology, Faculty of Science, Charles University, 12844 Prague, Czech Republic

**Keywords:** ER-PINs, PIN proteins, PIN5, PIN8, auxin transport, auxin metabolism

## Abstract

The proper distribution of the hormone auxin is essential for plant development. It is channeled by auxin efflux carriers of the PIN family, typically asymmetrically located on the plasma membrane (PM). Several studies demonstrated that some PIN transporters are also located at the endoplasmic reticulum (ER). From the PM-PINs, they differ in a shorter internal hydrophilic loop, which carries the most important structural features required for their subcellular localization, but their biological role is otherwise relatively poorly known. We discuss how ER-PINs take part in maintaining intracellular auxin homeostasis, possibly by modulating the internal levels of IAA; it seems that the exact identity of the metabolites downstream of ER-PINs is not entirely clear as well. We further review the current knowledge about their predicted structure, evolution and localization. Finally, we also summarize their role in plant development.

## 1. Introduction

The systematic effort of plant molecular biology has integrated the phytohormone auxin in key developmental processes. Auxin (in the most common form, indole-3-acetic acid, IAA) is closely linked with embryogenesis, organogenesis, tissue patterning, regeneration, tropisms [1], and also with a developmental response to stress [2] and other environmental stimuli [3]. Besides auxin signaling, which is located in the nucleus, and controlled IAA synthesis and degradation, which occurs in various cellular compartments, the critical role of the cell-to-cell auxin transport has been classically underlined [4]. In line with this concept, auxin is directionally conveyed by the PIN-FORMED family transporters (PINs) that are typically asymmetrically localized at the plasma membrane (PM) [5]. Besides the PM-located PINs and other auxin transporters present there [6,7,8], several studies also uncovered intracellularly located transporters WALLS ARE THIN 1 (WAT1) [9], and especially PIN-LIKEs (PILS) that show several structural and phenotypic features related to PINs and are localized on the membrane of the endoplasmic reticulum (ER) [10,11]. Importantly, there is a functionally and structurally distinct group of PINs located on ER as well [12,13,14,15].

The genome of *Arabidopsis thaliana* encodes eight PIN proteins. Based on their mutual sequence similarity and subcellular localization, they are categorized into two basic sub-clades (Figure 1): plasma membrane (PM- or long) and endoplasmic reticulum (ER- or short) PINs [12,16]. Well-characterized PM-PINs (PIN1, 2, 3, 4, and 7) mediate polar auxin efflux from the cells [4]. In addition, three ER-PINs (PIN5, 6, and 8) show a significant role in maintaining intracellular auxin distribution and homeostasis [12,13,14].

## 2. Membrane Topology and Structural Properties of ER-PINs

Based on the predictions of their membrane topology, PINs consist of three main structural units (Figure 2a): the N-terminal part, (central) hydrophilic loop (HL), and C-terminal part [19,20]. The N- and C-terminal parts contain each five alpha helices, with the properties of transmembrane domains (TMD), and these are interconnected with minor extra- and intracellular loops. In contrast to the N- and C-terminal parts, which show high conservation among all PINs, HLs show high heterogeneity in the amino acid composition and size [3,19,21]. While the usual length of HL of the *A. thaliana* PM-PINs varies between 301 and 336 amino acids, the HL of PIN5 and PIN8 contains 38 and 51 residues, respectively. PIN6 possesses HL of an intermediary length with 259 amino acids. In the HLs of PM-PINs, four conserved motifs termed HC1–HC4 have been identified. These motifs are missing in PIN5 and PIN8. The HL of PIN6 shows a partial homology with PM-PINs compared to that of PIN5 and PIN8. One or two of these motifs can be recognized in its sequence [22]. PIN6, despite the presence of these motifs, is more related in sequence to other short PINs [22] (Figure 1). Hence, among short PINs, PIN5 and PIN8 appear to be more structurally similar, while PIN6 shows several hallmarks also seen by long PINs.

## 3. The Structural Features Determining the Subcellular Localization of PINs with Short HL

The pronounced presence of PIN5 and PIN8 on ER was evidenced by their co-localization with ER markers and fluorescent ER-trackers in *A. thaliana* [12,13,14]. This localization pattern was observed in various tissues, even outside of their native expression domain [12,13,24,25]. Translational fusion of a short PIND from *Physcomitrium* (*Physcomitrella*) *patens* shows an intracellular localization as well [26]. Importantly, PIN6 protein resides at both PM and ER [14]. PIN6 was, besides ER, also co-localized with the Golgi and trans-Golgi network [14]. No comparable analysis has been conducted for PIN5 and PIN8, and it is also not entirely known which of these compartments can be indeed vital for the direct function of ER- or short PINs.

It was shown that ER-PINs displays low conservation around a tyrosine motif on the HL, predicted to be required for PM localization. PIN1-GFP carrying mutations in this region is retained in the ER [12]. The phosphorylation status of selected amino acid residues mediates various properties of PM-PINs [15,27,28,29]. Candidate phosphorylated residues were also found on the HL of PIN6 [30,31]. Replacing threonines T392 and T393 with glutamates, mimicking their constitutive phosphorylation, shifts the localization of PIN6 to PM [31]. These residues can be phosphorylated by mitogen-activated protein kinases (MAPKs) MPK4 and MPK6 in vitro. The authors [31] propose that PIN6 resides by default on ER when its expression level is low. The increased PIN6 expression, e.g., by specific developmental signals, can lead to the phosphorylation of PIN6 and its translocation to PM. These Ser/Thr phosphorylation sites are missing in PIN5, PIN8, and PM-PINs as well [31]. Moreover, when HL of PIN5 and PIN6 were replaced with the HL of PIN2, they showed PM localization [32,33]. It therefore seems that the sequence motifs present at the HL are required for the subcellular localization of the PIN proteins [12,32].

However, some studies suggest that other structural regions probably contain cues required for the localization of PINs. When the N- and C-terminal parts of PIN5 were swapped with those of PIN2, the chimeric protein displayed a dual PM and ER localization [33]. In addition, the expression of KfPIN from *Klebsormidium flaccidum*, which possess HL of an intermediate length [34], in *P. patens*, *Nicotiana tabacum* BY-2 tissue culture cells and *A. thaliana* seedlings lead to its PM localization [18]. PIN5-GFP originating from ramie (*Boehmeria nivea*) shows a PM localization when expressed in the leaves of relatively evolutionarily distant tobacco [35]. Ectopic expression of PIN8 in the pericycle [36] and root hair cells [27] results in localization at both ER and PM. In sum, shorter HL generally implies the ER location of PIN. However, other additional mechanisms are likely involved in determining the localization of PINs between the ER and PM.

## 4. Directionality of ER-PIN-Mediated Auxin Fluxes

The remarkable localization of ER-PINs argues for the importance of compartmentalization of internal auxin pools, where these proteins (along with other internal auxin transporters) mediate its transport between the cytoplasm and ER [12] (Figure 2b). *pin5* knock-outs and *PIN5* overexpressors show significantly increased and decreased IAA export, respectively. It was therefore inferred that PIN5 transports auxin from the cytoplasm into the ER lumen [12]. Notably, *pin8* knock-out and *PIN8* overexpressor showed opposite values in analogous experiments, suggesting that PIN8 may convey auxin from ER to cytoplasm, opposite to PIN5 [13]. Accordingly, the *PIN5* overexpressor shows a shorter hypocotyl [12], while the long one was observed by the *PIN8* overexpressor [13]. These overexpression effects were counterweighted in the double overexpression line, further evidencing mutually antagonistic roles of PIN5 and PIN8 [13]. The findings of the opposing role of the PIN5 and PIN8 were further independently confirmed by the analysis of auxin sensor intensity in protoplasts overexpressing *PIN5* and *PIN8* [37] and by examining the leaf vein phenotypes of the higher-order *ER-pin* mutants [25]. Due to the dual localization of PIN6 on PM and ER, the interpretation of the transport assays is challenging, and the direction of the PIN6-mediated flux at ER remains less clear. Free IAA levels in the *PIN6* overexpressor, however, resemble that of the *PIN8* overexpressor [13,14]. Altogether, it is assumed that PIN5 mediates the transport into the ER lumen, while PIN8 conveys auxin in the opposite direction; the PIN6-dependent auxin flux on ER perhaps resembles that of PIN8.

## 5. Evolution of ER-PINs in Plants

Because the long PINA from *P. patens* expressed in tobacco BY-2 culture cells displayed ER localization, it was formerly presumed that the reduced HL and the presence on ER of PINs might be evolutionarily ancestral [12]. Later, immunolocalization in situ and translational PINA-GFP fusions demonstrated that PINA localizes polarly at PM in the native *P. patens* tissues [26,38]. There is no closer similarity of HLs among different groups of short PINs; the members of each of these groups contain diverse sequence motifs [22]. A detailed phylogenetic analysis of PINs from *Embryophyta* [22] (simplified on Figure 3) suggests that short PINs, at least in angiosperms, evolved several times independently from long PINs by losing corresponding amino acid residues in the HL. Because the algal PINs identified seem to be present on PM [18,39], long, PM-localized PINs are probably the ancestral form of the PIN proteins in the seed plants [22,40]. Nevertheless, as the putative consensus sequence of the primeval PINs, including HL, is still not known, experimental validation of more, especially algal, genomic resources, is thereby further required [18,41].

## 6. Role of ER-PINs in Maintaining Intracellular Auxin Homeostasis

In contrast to the cell-to-cell transport mediated by PM-PINs, ER-PINs contribute to the distribution and availability of auxin in plants by modulating IAA metabolism [12,13]. The most common low-molecular products of IAA degradation are amino acid and sugar conjugates. Amino acid conjugates are converted reversibly, the other ones generally irreversibly [23]. IAA is also irreversibly oxidized to oxindole-3-acetic acid (oxIAA) [42]. *PIN5* overexpression in tobacco BY-2 cells leads to an increased ability to metabolize exogenously supplied IAA and its immediate IAA-glucose conversion product. *A. thaliana PIN5* overexpressors show elevated levels of IAA conjugates (IAA-aspartate and IAA-glutamate) and lower content of IAA [12]. The overall metabolic profiles of radioactive IAA were altered following induction of the *A. thaliana PIN8* gene in BY-2 cells as well. *A. thaliana PIN8* overexpressor shows decreased amounts of IAA-aspartate and IAA-glutamate, as well as oxIAA (oxIAA levels in *PIN5* overexpressors have not been tested) and also slightly increased levels of free IAA [13]. Increased levels of sugar-linked IAA were observed in tobacco plants overexpressing PIN8 from *A. thaliana* and the amounts of free IAA differed according to the tissue source in this transgene [43]. The quantities of both free IAA and IAA-Asp in *PIN6* overexpressor were higher than in the wild type and *pin6* mutant [14]. In sum, although the fate of IAA metabolites downstream of ER-PINs appears to be complex and a comprehensive analysis under comparable conditions is about to be done, it is apparent that ER-PINs are functionally upstream of pathways, leading to auxin deactivation (Figure 2b).

The *GRETCHEN HAGEN* (*GH3*) mRNAs have been initially identified among early auxin response transcripts [44]. Subgroup II of the broader GH3 family catalyzes the conjugation of IAA to amino acids [45,46]. Although most of them have been only little characterized, they appear to be usually localized in the cytoplasm [23,46,47] and probably function downstream of ER-PINs, even of those where the IAA degradation in ER is anticipated [12]. Di Mambro et al. [47] presented pieces of functional evidence that GH3.17 is a factor regulating root meristem development downstream of PIN5 (discussed below) (Figure 2b). A possible connection between *GH3s* and *ER-PINs* was documented by revealing that the transcriptional factor WRINKLED 1 (WRI1) binds to the promoters of *GH3.3*, *PIN5*, and *PIN6* (also to *PIN1* and *PIN4*). The expression of *GH3.3* and the *PIN* genes examined is altered, and the levels of IAA-Asp, but not IAA, are changed (elevated) in the *wri1* knock-out too [48,49].

It was shown that the ER stress sensor IRE1 is, together with PIN5, required for the activation of unfolded protein response. *ire1* mutants show auxin-related phenotypes, altered expression of auxin response genes, and also lower free IAA levels. The *ire1* defects were enhanced by the introduction of the *pin5* mutation into the *ire1* background. An elevated expression of the markers for the unfolded protein response was observed in several mutants involved in the ER-mediated auxin transport, regardless of the presumed defects in direction of the auxin flux [50]. In another study, it was proposed that the monovalent ion antiporters AtNHX5 and AtNHX6, which are localized at ER, mediate PIN5 function on ER, probably by changing pH in the ER lumen [51]. An interesting concept was proposed by Middleton and coauthors [37]. They observed that fluorescently labeled auxin enters the nucleus via ER. Based on the analysis of phenotypes and auxin sensors in protoplasts overexpressing PIN5 and PIN8, they propose the primary function of ER-PINs in tuning this process. These experiments were further corroborated by extensive mathematical modeling. They also predict the existence of unknown transporters mediating auxin flux between the ER lumen and nucleus [37] (Figure 2b). A follow-up investigation will likely explain the impact of ER-PIN on plant morphogenesis within this scheme.

## 7. Role of ER-PINs in Plant Development

ER-PINs are involved at various stages of root development and influence overall root growth [12,36,52]. *pin5* knock-outs show, among otherwise subtle phenotypes, slightly reduced primary root elongation and a lower initiation rate of lateral roots [12]. The expression of PIN5 in the root tip is barely detectable [12,36]. However, there is a distinct expression of the PIN5-GFP translational fusion in the vascular cylinder at the beginning of the primary root elongation zone [32]. Also, *pin8* knock-outs show a reduced lateral root density (which cannot be rescued by expression of *PIN5* under *PIN8* promoter), and the PIN8-GFP reporter is expressed in the phloem cells of the primary root tip stele [36]. In the *pin6* knock-outs, there are more lateral root primordia in the late stages of lateral root development compared with early stages, hinting at the participation of PIN6 in lateral root development as well [14,52]. The formation of adventitious roots is also perturbed in *pin6* knock-outs [14]. However, the role of other ER-PINs is unknown in this process.

Di Mambro et al. [47] reported that the *pin5-3* knock-down, however, shows an increased root growth and meristem cell number in their experimental setup. They also show that GH3.17-GFP is expressed in the lateral root cap. This coincides with data from their expression profiles, indicating that *PIN5* is enriched in the lateral root cap as well. The meristem cell number and root length in the *gh3.17*, as well in the *pin5-3 gh3.17* double mutants, were higher than that of the wild type. The *pin5* mutation was also able to revert the *GH3.17* overexpressor root meristem phenotypes. They therefore conclude that PIN5, together with GH3.17, may regulate auxin levels in the lateral root cap and thus meristem size and root growth. Expression of *GH3.17* is controlled by the cytokinin-dependent transcription factor ARR1 [53], and the intensity of auxin reporter seems to be unchanged following cytokinin treatment on the *pin5* and *gh3.17* mutants. It therefore seems that cytokinin may exert part of its effects on the root growth via this mechanism [47]. GH3.17-GFP is localized in the cytoplasm, while it is believed that the auxin degradation machinery downstream of PIN5 is present in the ER lumen [12]. Thus, the interaction between GH3.17 and PIN5 can be more complex, perhaps involving IAA conjugate hydrolases which have been identified at ER [54] or other IAA degradation enzymes and transporters [47] (Figure 2b).

In rice, it was found that suppressing the *PIN5B* expression by RNAi leads to altered panicle length, tiller number, yield, and other parameters of the above-ground plant growth. These lines also showed decreased levels of free IAA and elevated quantities of IAA-Asp, while the overexpressors showed concomitantly opposite values [55]. Notably, an intensive experimental effort also revealed that the leaf vein network in *A. thaliana* is mediated, parallel to the PIN1-dependent pathway, also by ER-PINs. *PIN5*, *PIN6*, and *PIN8* are expressed in leaf vascular founder cells [12,25,56]. The *pin6 pin8* double mutant displays a more complex vein network, while the single or the other double mutant combinations show a wild-type pattern. However, the observed complex vein phenotype of *pin6 pin8* is reverted in the *pin5 pin6 pin8* plants and becomes similar to the wild type [25]. Thus, this further supports the scenario that PIN5 and PIN8 (together with PIN6) function antagonistically on the ER membrane. It also demonstrates that, in contrast to that of PM-PINs [57,58], simultaneous disruption of all genes encoding ER-PINs leads to relatively mild phenotypes.

It has been earlier suggested that auxin transport plays a role also in pollen development [59]. PIN8 shows a pronounced expression in pollen, in contrast to PIN5 and PIN6 [13,43]. A high frequency of pollen defects, such as aborted and misshaped grains and reduced pollen germination ability, were observed in the *A. thaliana pin8* knock-outs and the *PIN8* RNAi lines in tomato [13,60], but not for the *pin5* and *pin6* mutants [13]. A loss of one or both short stamens was observed in the flowers of the *A. thaliana pin6* mutants [24]. Tomatoes carrying the *PIN8* RNAis show defects in the anther development in addition to seedless fruits observed [60]. Transcriptional profiling and reporter assays showed enriched expression of PIN6 in *A. thaliana* nectaries. A positive correlation was also found between the PIN6 expression and nectar production in *A. thaliana*. *pin6* mutants showed a decreased nectary size and incomplete expansion of petals. Moreover, following exogenous application of auxin or *N*-1-naphthylphthalamic acid (NPA, auxin transport inhibitor), *pin6* mutants did not show any significant changes in nectar production. In contrast, wild-type plants displayed increased and decreased nectar production after exogenous auxin and NPA treatment, respectively, further underlining the role of PIN6 in nectary development and generative organ development [24].

## 8. Conclusions

Auxin plays an indispensable role from the embryonic to the terminal stages of plant development. ER-PINs fine-tune auxin-dependent processes, mediating auxin fluxes over the membrane of the ER. The orientation on the membrane determines the direction of auxin transport between the ER and cytoplasm. However, the main and functionally relevant degradation products downstream of the ER-PIN activity should be further investigated. In addition, their relation to the functionally similar PILS proteins [10,11], which also reside on ER and show similar features in modulating auxin homeostasis, also appears as a prospective area in the research on the internal auxin homeostasis. Although genes encoding ER-PINs are present in all land plants, they show only relatively mild overall phenotypes in laboratory conditions. They can, therefore, perhaps be required for the interaction with the environment. Alternatively, they might be involved in an important but previously partly overlooked developmental process.

## Figures and Tables

**Figure 1 plants-09-01527-f001:**
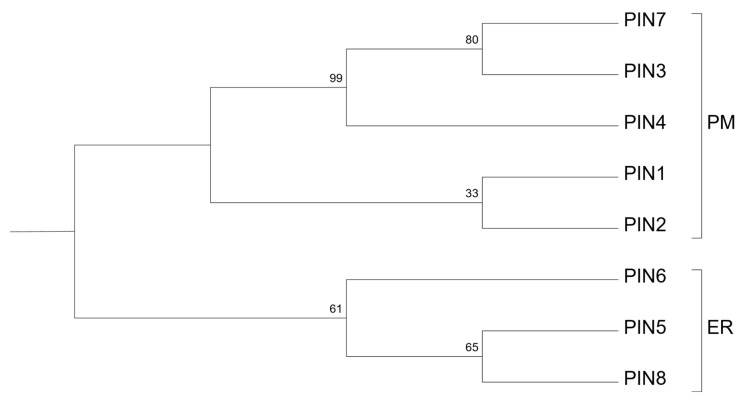
A ClustalW-based maximum likelihood [17] phylogenetic tree of the PIN proteins in *A. thaliana*, rooted with KfPIN [18]. PM: PINs present on PM, ER: PINs showing ER or dual PM and ER localization.

**Figure 2 plants-09-01527-f002:**
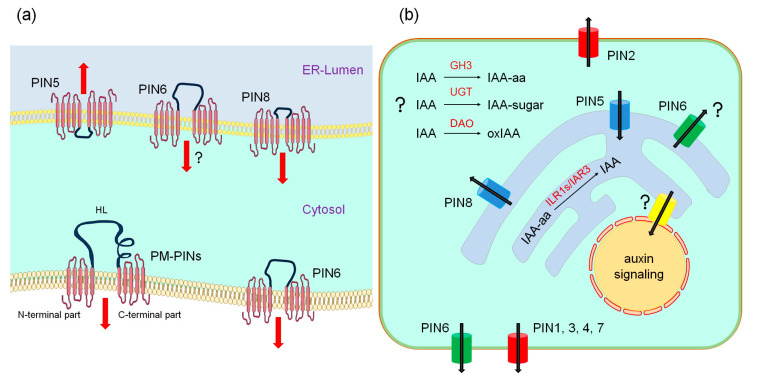
(**a**) The membrane topology, structural properties, and proposed direction of PIN-mediated auxin transport on PM and ER. The composition of the main structural units is illustrated on PM-PINs. (**b**) A scheme of the ER-PIN-mediated auxin transport, including the downstream IAA metabolism. Conversion of IAA to oxIAA or its conjugation to amino acid occurs most probably in the cytoplasm, while IAA de-conjugation was seen in the ER lumen. The subcellular localization of IAA conjugation with sugar moieties is unknown [23].

**Figure 3 plants-09-01527-f003:**
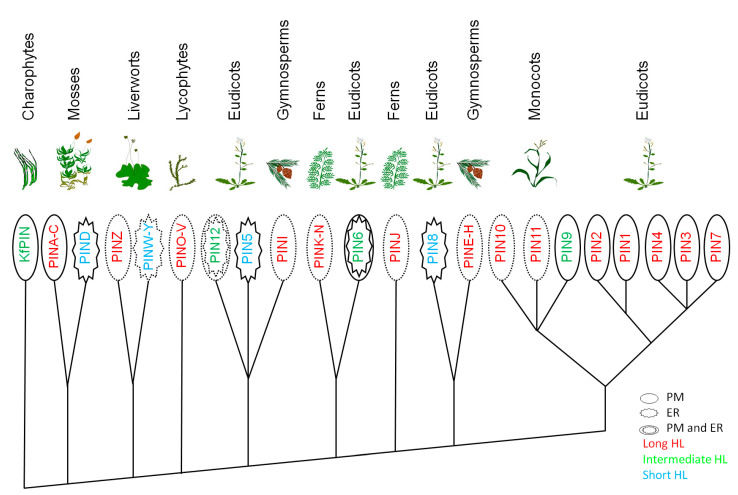
A simplified phylogenetic tree of the major PIN lineages in plants as proposed by Bennett et al. [22], including algal PIN from *Klebsormidium flaccidum* [18]. The taxonomical groups where the subcellular localization of the representative PIN transporters was experimentally confirmed are marked with a solid line, and the predicted localization is denoted by the dashed line.

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
