# Peer review of "ER-Localized PIN Carriers: Regulators of Intracellular Auxin Homeostasis"

_plants, 2020, doi:10.3390/plants9111527_

Round 1

Reviewer 1 Report

Although in the last years important reviews on PIN auxin transporters have been published, the review of Sisi and Ruzicka (plants-979355) manages to bring original insights into this topic. The authors focus their review on the endoplasmic reticulum (ER)-localized PIN proteins, which was a good idea, considering that the scientific community still lacks a comprehensive view regarding the role of auxin transporters/facilitators at the ER. The authors discuss the findings and the hypothesis related to the ER-PIN proteins. Although very focused, this topic is actually of interest to a large audience interested in cellular auxin homeostasis, making this review a gain for the scientific literature.

Minor points:

-I would rather use in the title (r4) “homeostasis or distribution” than “streams”. We know more about auxin homeostasis and less about the auxin streams regulated by ER-PINs.

-r10: In this context, I would call auxin “hormone” and not “substance”.

-r18: I would end the abstract with a more positive conclusion or a conclusion centered on what the review is discussing (instead “it seems that the metabolic processes ....).

-I find the insertion of the chapter 5. (about evolution) between the chapters 4. and 6. (describing the function of ER-PINs) disturbing the flow of the information. I would suggest to start with the evolution of ER-PINs right after the Introduction or after Chapter 2. (Structural properties) and then let the MS flow with the other chapters. Of course, this is a matter of taste, and not necessarily needed.

-Figure 2, b: sugar instead suger

-r161: replace “both” with a correct term

-r166: .... “and” lower free IAA level.

Reviewer 2 Report

The manuscript by Sisi and Růžička, is well organized and focused. There are only a couple of places where the English sentence framing needs some work (e.g. line 224-225), but overall the language is quite lucid and easy to understand. To the best of my knowledge, there are almost no reviews analyzing the role of the ER PINs in auxin signaling and transport and this manuscript actually does a great job in summarizing the data from all the relevant articles and provide a wide perspective on the function ER-PINs.

Suggestions:

  1. The authors do mention in places about the dual localization pattern of the ER-PINs (PIN5, 6 and8). However, there is not much discussion around the point as to why the change of localization can be significant. The data from multiple publications suggests that the change of localization is either cell-type specific or developmentally regulated. I would encourage the authors to discuss it in a more focused manner. One of the major and proven finding in PIN biology is that PIN localization can change even for PM-PINs depending on the tissue type they are expressed in to perform cell-type specific auxin transport, so that can apply to ER-PINs also.
  2. I failed see much reference to the PIN-LIKEs (PILs) proteins and there role in ER auxin homeostasis. Can the authors also include a few lines on that and how PILs can act together with ER-PINs in auxin transport?
  3. Reference 33 and 34 are repeats.
  4. Line 10 (abstract), please change the word ‘substance’ to either molecule or hormone or regulator (or anything more defined).
